# Poly(ADP) ribose polymerase promotes DNA polymerase theta-mediated end joining by activation of end resection

Megan E. Luedeman [1], Susanna Stroik[2], Wanjuan Feng[2], Adam J. Luthman[3], Gaorav P. Gupta [1,2,3,4] & Dale A. Ramsden [1,2,3] ✉

The DNA polymerase theta (Polθ)-mediated end joining (TMEJ) pathway for repair of chromosomal double strand breaks (DSBs) is essential in cells deficient in other DSB repair pathways, including hereditary breast cancers defective in homologous recombination. Strand-break activated poly(ADP) ribose polymerase 1 (PARP1) has been implicated in TMEJ, but the modest specificity of existing TMEJ assays means the extent of effect and the mechanism behind it remain unclear. We describe here a series of TMEJ assays with improved specificity and show ablation of PARP activity reduces TMEJ activity 2-4-fold. The reduction in TMEJ is attributable to a reduction in the 5′ to 3′ resection of DSB ends that is essential for engagement of this pathway and is compensated by increased repair by the nonhomologous-end joining pathway. This limited role for PARP activity in TMEJ helps better rationalize the combined employment of inhibitors of PARP and Polθ in cancer therapy.

Double-strand breaks (DSBs) are repaired in mammalian cells via homologous recombination (HR), nonhomologous-end joining (NHEJ), or the still poorly understood "alternative"-end joining pathway(s) (a-EJ). Deficiencies in DSB repair lead to genome instability and cell death, and eventually, cancer or developmental disease at the organismal level[1]. Of particular importance, defects in genes important for HR (e.g. *BRCA1/2*) account for the majority of hereditary breast, ovarian, and prostate cancers[2]. Dysregulation of repair pathway choice can also contribute to disease. The choice of repair pathway is partly determined by end resection, a progressive 5′ to 3′ degradation of one strand of each end of the DSB that generates 3′ overhanging single-stranded DNA (ssDNA) tails[1]. Resection impedes repair by NHEJ and enables repair by both HR and a-EJ[3].

It remains difficult to accurately assess repair by a-EJ pathway(s)[4]. a-EJ has historically been defined as DSB repair independent of factors required by either NHEJ and HR and is associated with an enrichment in repair products with deletions extending to sequence identities in DNA flanking the DSB (microhomologies)[5–7]. At least in mammals, the majority of repair typically associated with a-EJ utilizes micro-homologies of 2–6 base pairs (bp) and is dependent on DNA polymerase theta (Polθ, encoded by *Polq*)[4,8–10]. Nevertheless, a-EJ can be detected independent of Polθ; such repair typically requires *Helq* and longer microhomologies[4,11,12]. Here we define the subset of mammalian a-EJ missing in *Polq*-deficient cells as theta-mediated end joining (TMEJ), to distinguish it from the still unclear contribution of theta-independent a-EJ[4].

Resected end intermediates impair NHEJ but are essential for both TMEJ and HR pathways; therefore, HR-defective cells are sensitive to deficiency in, or inhibition of, Polθ (Polθi)[13–16]. The "synthetic lethality" of combined defects in HR and a second DNA repair pathway is also observed with inhibitors of poly(ADP) ribose polymerase 1 (PARP1 inhibitors; PARPi)[17,18]. Consequently, PARPi and Polθi are both being employed as therapies to target HR defective cancers. PARP1 is activated by binding to strand breaks, and it then modifies interacting proteins and itself with chains of poly(ADP) ribose (PARylation)[19]. PARPi is thought to result in dysfunctions in genome replication

[1]Curriculum in Genetics and Molecular Biology, University of North Carolina at Chapel Hill, Chapel Hill, NC, USA. [2]Lineberger Comprehensive Cancer Center, University of North Carolina at Chapel Hill, Chapel Hill, NC, USA. [3]Department of Biochemistry and Biophysics, University of North Carolina at Chapel Hill, Chapel Hill, NC, USA. [4]Department of Radiation Oncology, University of North Carolina at Chapel Hill, Chapel Hill, NC, USA. ✉e-mail: dale_ramsden@med.unc.edu

specifically toxic to HR-defective cancers[20–23]. The mechanisms for sensitization of HR-defective cancers to PARPi vs. Polθi are, thus, at least partly distinct, which is consistent with the additive cytotoxicity of these two inhibitors observed in HR defective cells[13,15,16].

However, interpretation of the additive effect of PARPi and Polθi is complicated by early work showing a-EJ is impaired by PARPi or deficiency in PARP1[24,25]. This role for PARP1 activity is likely unrelated to the role of PARPi in causing replication dysfunction. It seems probable that PARP deficiency or inhibition only partly impairs repair by TMEJ, or the effects of PARP1 on repair by a-EJ are wholly or partly attributable to a role for PARP1 in theta-independent a-EJ. Relevant to this latter point, inhibition of PARylation impairs recruitment of Polθ to DNA damage in cells[14,26], and PARylation of the N-terminal domain of Polθ promotes its dissociation from DNA in vitro[27], but it is unclear if either of these observations is sufficient to result in significant deficiency in cellular TMEJ.

Attempts to clarify the role of PARP1 in cellular TMEJ have been undercut by difficulties in unambiguously quantifying the extent of repair mediated by this pathway vs. NHEJ and theta-independent a-EJ. To resolve this concern, we developed extrachromosomal and chromosomal assays that measure TMEJ with high specificity: levels of repair detected using these assays were more than 10-fold higher in wild-type cells relative to isogenic *Polq-/-* cells. We found that deficiency in PARP signaling had no impact on TMEJ as assessed using extrachromosomal, "pre-resected" substrates. By comparison, levels of chromosomal TMEJ were reduced two to four-fold, whether using high levels of PARP inhibitor or cells deficient in PARP1 and its candidate backup, PARP2. This reduction of TMEJ levels in PARP-deficient or PARPi-treated cells paralleled decreased damage-dependent colocalization of Polθ with the resection factor CtIP, reduced levels of end resection, and finally, compensatory increases in repair by NHEJ. Our data are consistent with an argument that PARylation does not directly impact steps specific to the TMEJ pathway. Instead, PARylation indirectly promotes Polθ recruitment and TMEJ activity at breaks by increasing the frequency of end resection and, thus, redirecting repair of these ends from NHEJ to TMEJ.

## Results

### Impact of PARPi on extrachromosomal and chromosomal TMEJ
The specific assessment of TMEJ activity is complicated by infrequent employment of this pathway in NHEJ proficient cells, and because the microhomology-mediated deletion products typically used as a surrogate for TMEJ are also preferential products of repair by NHEJ and theta-independent a-EJ[4]. We previously described an extrachromosomal substrate assay that emphasized repair by TMEJ, relative to NHEJ or theta-independent a-EJ, by introducing into cells linear DNA fragments with >45 nucleotide (nt) ssDNA 3'-overhangs (thus "pre-resected") with a short (4 bp) microhomology. We then exclude a contribution of NHEJ after loss of the 3' overhangs by quantifying only those repair products that retain the ssDNA overhang sequences[9,10]. Here, we additionally alter the end structure to include unpaired 5' overhangs with sequence that directs formation of an intramolecular hairpin (Fig. 1a), which helps ensure repair requires synthesis directed from 3' ssDNA tails. We also account for differences in efficiency of substrate introduction into cells by inclusion of a second extrachromosomal substrate that measures repair by NHEJ (a "spike-in control"). Repair products of these TMEJ and NHEJ substrates are then recovered from cells and measured in parallel, in a single multiplexed quantitative PCR reaction (qPCR), using 5'-nuclease probes that are specific to products of the two different repair pathways (Supplementary Fig. 1a, b). We further confirmed repair measured by the spike-in control was independent of *Polq* deficiency and PARP inhibition (10 μM of olaparib) (Supplementary Fig. 1c). We performed this assay in SV40 T-antigen transformed mouse embryonic fibroblast cell lines (MEFs), comparing wild-type cells (WT) to cells derived from mice

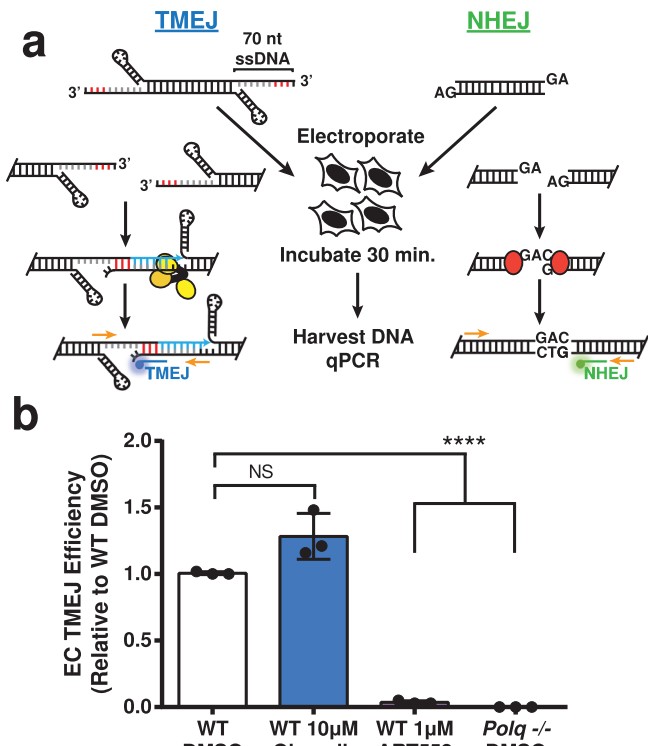

**Fig. 1 | PARPi does not impact extrachromosomal TMEJ. a** Schematic of the extrachromosomal assay (EC) and pathway-specific probe qPCR (orange primers with blue TMEJ and green NHEJ probes). The microhomology in the TMEJ substrate is shown in red and Polq-dependent synthesis in blue. The helicase-like and polymerase domains of Polθ are shown as yellow ovals, and Ku is shown as red ovals. **b** TMEJ substrate joining efficiency, normalized with NHEJ substrate joining efficiency to account for differences in transfection efficiency, is expressed as a fraction of the TMEJ observed in wild-type MEFs treated with vehicle (WT DMSO). Data shown are mean ± standard deviation (SD) (*n* = 3). Statistical significance of differences, relative to WT DMSO cells, was determined by one-way ANOVA (****$p < 0.0001$; NS, $p = 0.87$, not significant).

deficient in *Polq* (*Polq-/-*). Repair of the improved TMEJ substrate was not detectable in *Polq-/-* cells (Fig. 1b) and reduced 20-fold following pre-treatment of cells with 1 μM of the Polθ inhibitor ART558 (Polθi). Serial dilution of products recovered from wild-type cells determined the limit of detection was <1% (>100-fold signal:noise).

We investigated the role of PARP activity in TMEJ by treating WT MEFs with vehicle (dimethyl sulfoxide; DMSO) or 10 μM of the PARP inhibitor olaparib, followed by introduction of the TMEJ substrate and the NHEJ spike-in control. TMEJ activity was measured as described above. Although this high dose of olaparib is sufficient to ablate all PARP activity[21], we see no evidence for an effect on the efficiency of TMEJ using this assay (Fig. 1b).

We considered next the possibility that TMEJ in the extrachromosomal assay is independent of PARP activity because PARP activity may have a role specific to chromosomal repair, such as chromatin remodeling or end resection. To address these possibilities, we employed a previously described chromosomal TMEJ assay, where a Cas9 nuclease guided by associated RNA generated chromosome breaks at a site in the *Rosa26* locus, after which qPCR was used to measure a microhomology-associated deletion (MHD) that was specifically depleted in *Polq*-deficient cells (R26^MHD, Fig. 2a, Supplementary Fig. 2a). In accord with previous work[9,10,28], R26^MHD is reduced 4-fold in *Polq*-deficient contexts (either *Polq-/-* cells, or in cells pre-treated with Polθi) (Fig. 2b), relative to wild-type controls. R26^MHD is similarly reduced upon treatment with PARPi and to a lesser extent in cells deficient in PARP1 and PARP2 (Fig. 2c, Supplementary Fig. 2b). This is

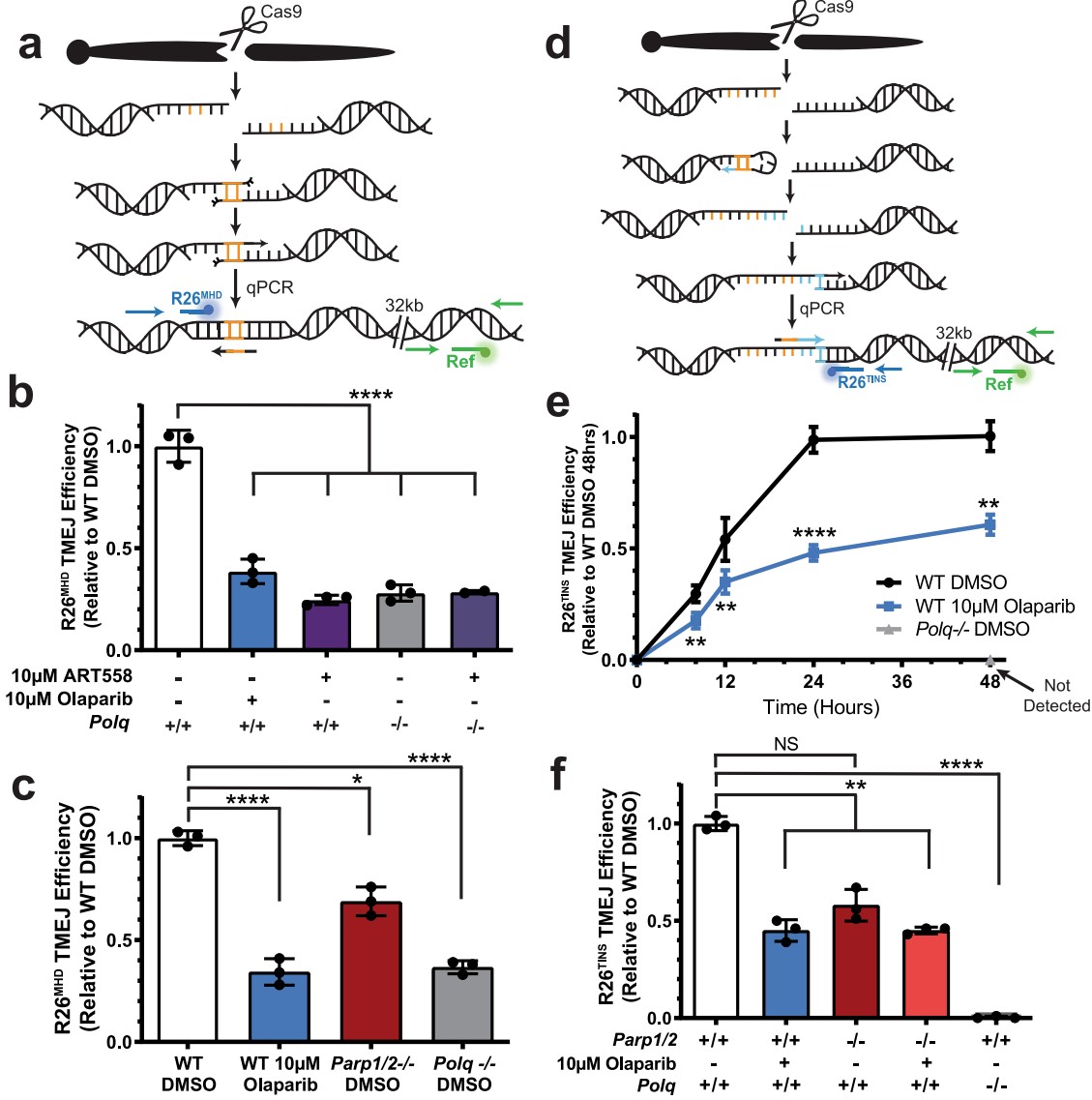

**Fig. 2 | Loss of PARP activity partially inhibits chromosomal TMEJ. a** Model of the formation of the microhomology (orange) mediated deletion signature TMEJ repair product (R26^MHD) and its detection by qPCR. The location of primers and probe is identified for the R26^MHD amplicon in blue and the reference amplicon in green. **b, c** R26^MHD TMEJ efficiency after 24 h comparing DMSO-treated WT or *Polq-/-* MEFs to MEFs treated with olaparib or ART558 (**b**, ****$p < 0.0001$) or *Parp1/2-/-* MEFs (**c**, *$p = 0.014$, ****$p < 0.0001$). **d** Model of the formation of the templated insertion (blue section) signature TMEJ repair product (R26^TINS) and its detection by qPCR. The location of primers and probe is identified for the R26^TINS amplicon in blue and the reference amplicon in green. **e** Accumulation of the R26^TINS TMEJ signature over 48 h in WT MEFs treated with DMSO (black) or 10 μM olaparib (blue; **$p < 0.0075$, ****$p < 0.0001$). *Polq-/-* MEFs had no detectable signal (grey). **f** R26^TINS TMEJ efficiency after 48 h in WT, *Parp1/2-/-*, and *Polq-/-* MEFs treated with DMSO or 10 μM olaparib (NS, $p = 0.067$, not significant; **$p < 0.01$, ****$p < 0.0001$). All signatures are normalized to the number of genomes as determined by a reference (Ref) amplicon 32 kilobases (kb) downstream and are expressed as a fraction of that observed in WT cells treated with DMSO. All data shown are mean ± SD ($n = 3$), and statistical significance of differences, relative to WT DMSO cells, was determined by one-way ANOVA.

consistent with a role for PARP activity in chromosomal TMEJ. However, the high background of this assay means it is not possible to definitively determine the extent of overlap between lost repair attributable to *Polq* deficiency (i.e. definitive TMEJ) and lost repair attributable to deficiency in PARylation.

We developed an assay for chromosomal TMEJ activity with greater specificity than that described above by focusing on a different class of chromosomal repair products: templated insertions (TINS). TINS are nearly unique to TMEJ, though typically less frequent and more heterogeneous than MHD products[10,29]. However, prior work from our group characterized repair at another site in the *Rosa26* locus depleted of nearby microhomologies, and consequently, TINS account for a much higher than typical fraction of TMEJ and total repair[10]. We used a qPCR designed to specifically detect TINS products at this site

(Fig. 2d, Supplementary Fig. 2d; R26^TINS) and confirmed they are not detectable in *Polq-/-* cells and present at over 10-fold higher levels in isogenic *Polq*-proficient cells (Fig. 2e). Compared to cells treated with vehicle, cells treated with 10 μM olaparib led to delayed accumulation of R26^TINS, eventually leading to a 39 ± 5% reduction in accumulation of these products. A comparable reduction in R26^TINS was observed in cells deficient in both PARP1 and PARP2 (Fig. 2f). Importantly, because R26^TINS is entirely *Polq* dependent, our data indicate loss of PARylation impairs, but does not ablate, chromosomal TMEJ. In sum, TMEJ on a pre-resected extrachromosomal substrate is fully independent of PARylation, while PARylation activity promotes, but is not absolutely essential for, TMEJ in chromosomal contexts.

These observations are consistent with data arguing there is additive sensitivity of HR defective cells to PARPi and *Polq* loss[13,15,16]

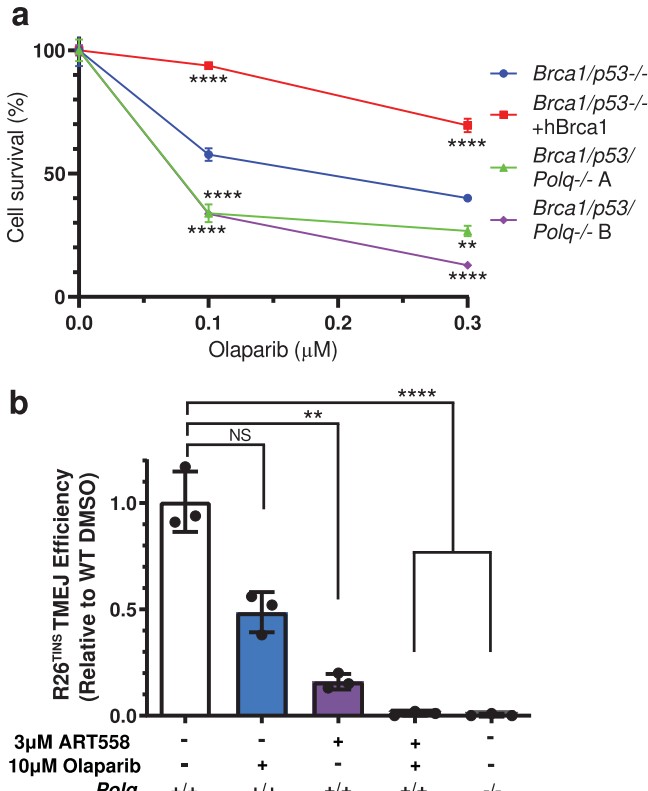

**Fig. 3 | Effects of combined impairment of TMEJ and PARP. a** Olaparib sensitivity of murine mammary tumor line KPB13 assessed by clonogenic survival assay. Parental line was *Brca1/p53-/-* (**$p = 0.0018$, ****$p < 0.0001$). **b** R26^MHD TMEJ efficiency after 48 h in MEFs treated with DMSO, olaparib, and/or ART558 normalized to WT DMSO (NS, $p = 0.23$, not significant; **$p = 0.0023$, ****$p < 0.0001$). All data shown are mean ± SD ($n = 3$). Statistical significance of differences, relative to the parental line (**a**) or WT DMSO (**b**) was determined by one-way ANOVA.

(Fig. 3a). An additive effect on TMEJ is also evident when using the R26^TINS assay in cells treated with sub-saturating levels of Polθi and saturating levels of PARPi (Fig. 3b).

**Mechanism for PARP's involvement in DSB repair/TMEJ**

The disparate results observed when comparing pre-resected ends (extrachromosomal assay, Fig. 1) vs. the near blunt Cas9-generated ends that require resection for TMEJ (chromosomal assays, Fig. 2) suggests the effect of PARP activity might be confined to the resection step. In agreement with previous work[14,26], inhibition of PARP reduced the frequency of Polθ foci that form after ionizing radiation (Fig. 4a, b). Here, we also observed PARP inhibition reduced the extent Polθ colocalized with an activator of resection, CTBP interacting protein (CtIP) (Mander's overlap coefficient reduced from 0.227 in vehicle-treated cells to 0.166 after treatment of cells with olaparib) (Fig. 4c, d).

We sought to directly assess if PARP inhibition affected end resection. We again introduced a targeted DSB at the *Rosa26* locus and recovered genomic DNA at various timepoints after breakage. We assessed resection near this DSB using droplet digital PCR (ddPCR), comparing mock-treated samples vs. samples treated with thermolabile Exo I, an exonuclease that specifically degrades the 3′ overhanging ssDNA tails generated by end resection. As expected, resection as detected by this assay is reduced in cells defective in Mre11 (*Mre11^ATLD1*; Supplementary Fig. 4a), and results are comparable to a previously described assay that detects ssDNA by its resistance to a restriction enzyme that cuts double stranded DNA (compare Supplementary Fig. 4b–c). 9.1 ± 0.3% of chromosomes had a minimum of 8 nt resected, relative to the break site, 4 h after break induction, and the

frequency of ssDNA at this site declined to 2.4 ± 0.8% by 24 h (Fig. 5b solid black line). Olaparib treatment significantly reduced resection as measured using this break-proximal site – it peaked at only 5.2 ± 0.8% of chromosomes – and the accumulation of these resected ends was slightly delayed (Fig. 5b, solid blue line). A similar impairment of short-range resection was also observed in cells deficient in PARP1 and PARP2 (Supplementary Fig. 4c). We additionally assessed resection further away from the break. A maximum of 3.8 ± 0.4% of chromosomes underwent resection up to 284 nt from the break (Supplementary Fig. 4d), and 1.7 ± 0.5% of chromosomes had ssDNA 527 nt from the break site (Fig. 5b dashed lines). While PARPi also impaired resection as assessed at these more distal locations, the extent of impairment was less pronounced and insufficient to explain PARPi-dependent effects on TMEJ. Finally, the early reduction in resected ssDNA ends was not compensated by increased chromosomes with either dsDNA broken ends (Supplementary Fig. 4e) or deletion of both strands (Supplementary Fig. 4f).

In sum, significant effects of PARP inhibition appear confined to an impairment of short-range resection and repair by TMEJ, and apparent re-channeling to another repair fate. Consistent with this argument, an NHEJ-specific signature repair product at this locus (an insertion of a single nucleotide; reduced 50-fold in Ku-deficient cells[28]) was enriched in cells treated with olaparib (Fig. 5d, Supplementary Fig. 2c). However, it is not possible to employ signature products to accurately estimate whether compensation is complete; thus, we sequenced all repair involving insertions and deletions (indels). We classified the TMEJ fraction as all deletions significantly reduced in *Polq*−/− cells[10] and the NHEJ fraction as all indels less than 5 bp without microhomology 2 bp or more[28] (Fig. 6a). Treatment of cells with olaparib led to reductions in the TMEJ fraction (−4.5 ± 0.9% relative to DMSO-treated WT MEFs) that were readily accounted for by compensating increases in repair by NHEJ (4.2 ± 1.5% relative to DMSO-treated WT MEFs; Fig. 6b).

Resection generates intermediates for both TMEJ and homologous recombination (HR). We used a gene targeting assay to determine if PARPi similarly reduces repair by HR. We repeated the experiment described above, except included a donor plasmid with homology flanking the Cas9 target site such that Rad51-dependent HR using this donor introduces a ScaI site (Fig. 6c)[9]. As observed with our chromosomal TMEJ assays, gene targeting was reduced 2-fold in olaparib-treated MEFs compared to vehicle-treated cells. Our data indicate PARP activity promotes resection and, consequently, repair by TMEJ and HR; in the absence of PARP activity, these breaks are re-channeled to repair by NHEJ (Fig. 6e).

## Discussion

Post-translational modification of proteins near strand breaks with PAR chains by PARP1 and PARP2 has diverse effects on the DNA damage response[19]. Relevant to this work, prior studies identified damage-dependent PARylation as an important promoter of the a-EJ pathway for repair of chromosome breaks[24,25]. Here, we show both inhibition of PARylation and deficiency in PARP1 and PARP2 impair end resection to a degree sufficient to explain the observed re-channeling of DSB repair from TMEJ and HR to NHEJ (Fig. 6e). This is consistent with evidence that recruitment of Mre11, a factor important for resection, to DNA damage is impaired in the absence of PARylation,[30,31] as well as our demonstration that inhibition of PARylation reduces ionizing-radiation-dependent colocalization of CtIP (also required for resection) and Polθ (Fig. 4d). We also see no evidence for effects of PARylation on TMEJ when this pathway is assessed independent of resection (i.e., in repair of pre-resected ends; Fig. 1). The PARP-dependent effects on resection and channeling to TMEJ observed here is consistent with prior observations that recruitment of PARP1 and the NHEJ factor Ku are mutually exclusive[25,31,32].

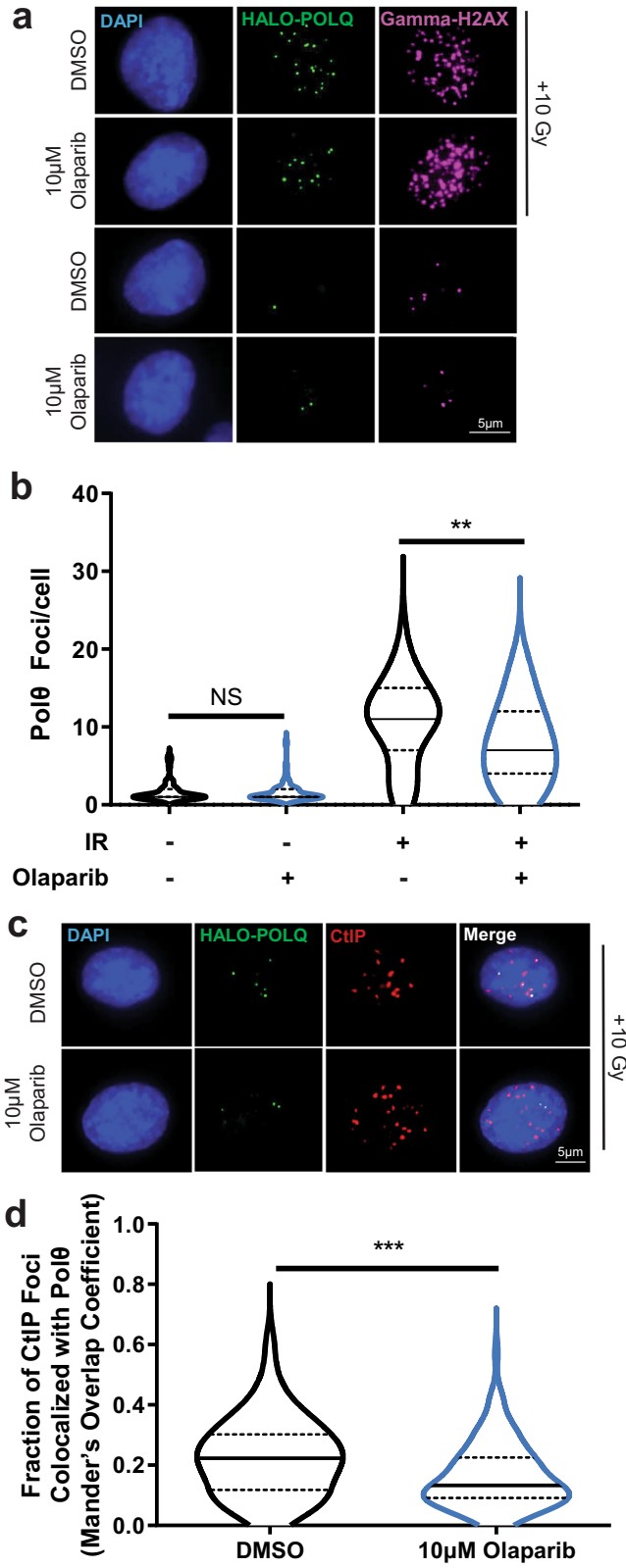

**Fig. 4 | PARP activity promotes Polθ foci formation and colocalization with CtIP. a** Representative images, selected from 300 imaged cells, of HALO-POLQ foci (green) in DMSO- or olaparib-treated RPE-1 cells without damage or with damage induced by 10 Gray (Gy) of ionizing radiation (IR). DAPI (blue) marks nuclei, and GammaH2AX (pink) indicates damage. **b** Number of Polθ foci per cell in cells with >0 foci (DMSO + IR $n = 55$, olaparib+IR $n = 43$, DMSO $n = 50$, olaparib $n = 50$). Significance of difference determined by one-way ANOVA (NS, $p > 0.99$, not significant; **$p = 0.0012$). The distribution of foci/cell is represented by a truncated violin plot including median (solid line) and upper and lower quartile (dashed lines) values. **c** Representative images, selected from 100 imaged cells, of HALO-POLQ (green) and CtIP (red) foci in irradiated RPE-1 cells. **d** Fraction of CtIP foci colocalized with Polθ (Mander's overlap coefficients; DMSO $n = 100$, olaparib $n = 100$). Significance of difference determined by unpaired, two-tailed t-test (***$p = 0.0003$). The distribution of foci/cell is represented by a truncated violin plot including median (solid line) and upper and lower quartiles (dashed lines) values.

The specific toxicity of PARPi in HR defective cancers is primarily attributed to the ability of PARPi to generate replication fork dysfunctions that preferentially engage HR[20–23]. Our data suggest the ability of PARPi to impair resection also exacerbates the original HR defect, providing an independent mechanism for specific toxicity (Fig. 6d, e), though note the inhibitory effect of PARPi on repair by HR is at best modest, and not similarly observed by others in a different model[38].

Taken together, our data are consistent with a model in which PARP1 instead of Ku is recruited to a minor subset of DSBs. The resulting activation of local PARylation facilitates (i) recruitment of factors (Mre11, Rad50, Nbs1, CtIP) important for short-range resection, (ii) generation of the 20–200 nt, 3' ssDNA overhangs that are preferential substrates for Polθ/TMEJ, (iii) recruitment of Polθ, and (iv) repair by TMEJ instead of NHEJ (Fig. 6e). It will be important to clarify in future work how PARylation effects resection, and recruitment of Mre11. We note the observed PAR-mediated increased recruitment of Polθ at breaks[14,26] (Fig. 4b) may be an indirect consequence of PAR-dependent effects on resection, possibly assisted by PAR-dependent unloading of replication protein A[39], a ssDNA binding protein that has been shown to inhibit a-EJ[40]. Additionally, the lack of a role for PARylation observed when TMEJ is measured using extrachromosomal, pre-resected substrates argues against a significant direct role for PARylation in promoting TMEJ after resection has been initiated. Of note, confinement of the role of PARPi to impairment of resection is consistent with similar inhibitor effects of PARPi on repair by HR (Fig. 6c, d), and suggests it could also inhibit Theta-independent a-EJ (Fig. 6e).

Resection and TMEJ are reduced to a greater degree when cells are treated with PARP inhibitors, relative to that observed in PARP deficient cells, though this difference is typically modest and not always statistically significant. The ability of PARP inhibitors to delay dis-association of PARP1 at ends ("trapping")[21] may interfere with the ability of the resection machinery to access DNA ends, eventually leading to a redirection of the repair of these ends to NHEJ.

TMEJ was only modestly impaired (reduced 2-4-fold) by deficiency in both PARP1 and PARP2, as well as by treatment of cells with saturating levels of PARPi. The remaining TMEJ activity in PARP inhibited cells was also sensitive to treatment with Polθi even when using sub-saturating concentrations of Polθi. This incomplete epistasis – the observation that even fully ablated PARylation activity only modestly impairs TMEJ activity – helps explain why the toxicity of PARPi and Polθi is additive in HR-deficient cancers, and better rationalizes the utility of combined therapy.

## Methods

### Materials

Olaparib was purchased from Selleck Chemicals, ART558 was provided by Artios Pharma Limited, and both were dissolved in dimethyl sulfoxide (DMSO, Sigma).

Prior evidence on the role of PARP activity in regulation of long-range resection, as well as recruitment of factors required for long-range resection (EXO1) has been contradictory[33–37]. Here we observe that levels of long-range resection (>284 nucleotides), as well as the effects of PARP inhibition on long-range resection (mildly inhibitory) are of insufficient magnitude to explain the influence of PARPi on DSB repair pathway choice.

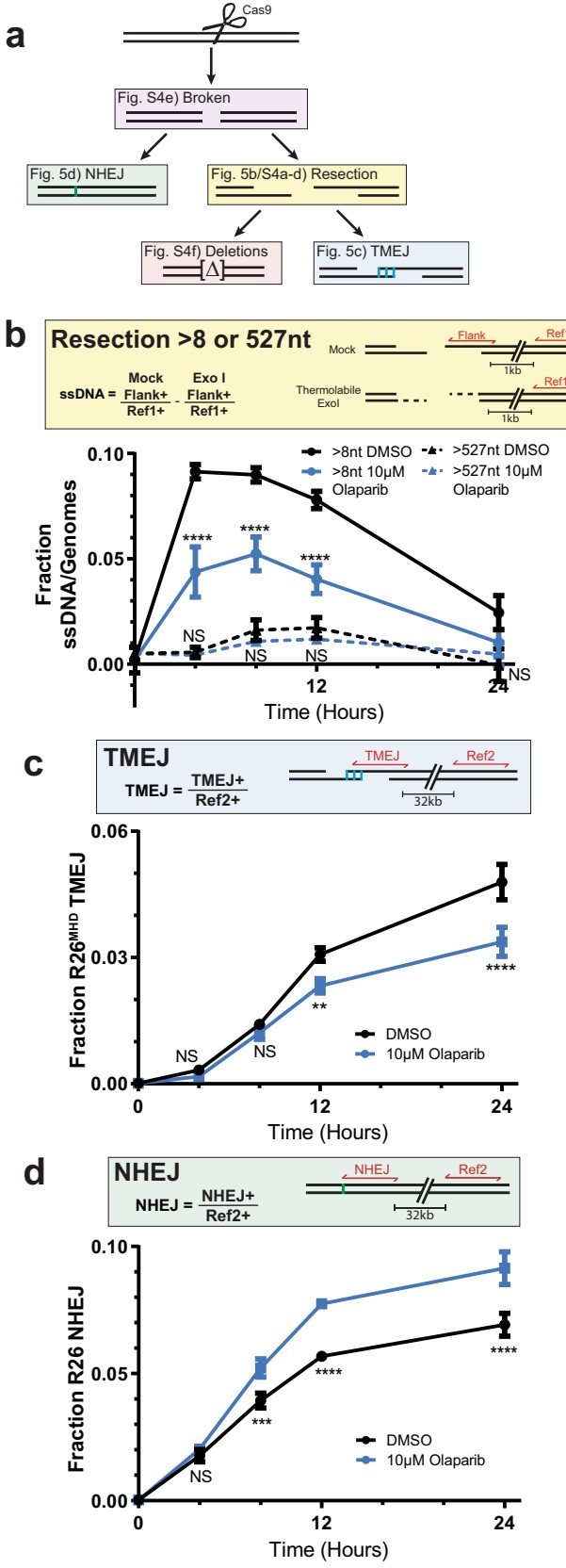

**Fig. 5 | Loss of PARP activity inhibits resection and promotes NHEJ. a** Diagram depicting repair intermediates and outcomes of a Cas9-induced DSB. Amplicons used for ddPCR are shown in red in the corresponding, color-matched boxes. The number of genomes in each PCR was determined by a reference amplicon -1 kb (Ref1) or -32 kb (Ref2) from the cut site. **b** Fraction of genomes with ssDNA >8 nt (solid lines) or >527 nt (dashed lines) generated by 5′-to-3′ resection in WT MEFs treated with DMSO (black) or 10 μM olaparib (blue) over 24 h. **c** Fraction of genomes repaired by R26$^{MHD}$ TMEJ signature or **d** R26 NHEJ signature in WT MEFs treated with DMSO (black) or 10 μM olaparib (blue) over 24 h. Data shown are mean ± SD of one biological replicate and four (B) or three (C and D) technical replicates; the same biological replicate was used in B–D. Statistical significance of differences, relative to WT DMSO cells, was determined by one-way ANOVA (*$p = 0.04$; **$p = 0.003$; ***$p = 0.0006$; ****$p < 0.0001$; NS, $p > 0.7$, not significant).

Sciences, Seradigm) and penicillin (5 U/mL, Sigma). MEFs were SV40 T-antigen transformed cells derived from mice proficient (wild type; WT) or deficient in *Polq* (*Polq−/−*) and were previously characterized[8]. Variant clones of the WT MEFs deficient in *Parp1* (*Parp1−/−*) were generated using Cas9 and the guide listed in Supplementary Table 1; *Parp2* deficiency was generated in a *Parp1−/−* clonal line by a subsequent introduction of Cas9 and the guide listed in Supplementary Table 1 to generate a clonal line deficient in *Parp1* and *Parp2* (*Parp1/2−/−*). *Mre11*$^{ATLD1}$ MEFs were generated by bi-allelic knock-in and previously characterized[28]. Retinal pigment epithelial (RPE-1) cells immortalized by human telomerase reverse transcriptase (hTERT) expression were cultured in DMEM F12 (Invitrogen) containing 10% Fetal Bovine Serum (VWR Life Sciences, Seradigm) and penicillin (5 U/mL, Sigma). A clonal HALO-tagged POLQ line was derived from retroviral transduction of a HALO-POLQ construct (a gift from Richard Wood). *Brca1/p53* null KPB13 murine mammary tumor cells were previously characterized[41,42] and plated in HuMEC with 5% Fetal Bovine Serum (VWR Life Sciences, Seradigm), penicillin (5 U/mL, Sigma), and the HuMEC (Gibco) supplementary kit (Gibco). A clonal line stably expressing human *BRCA1* was generated by transfer of the cDNA from Addgene 52504 to pEZY3 (Addgene #18672), linearization, and selection for clones with expression as confirmed by RT-qPCR. Variant clones of the parental KPB13 line deficient in *Polq* were generated using Cas9 and the guides listed in Supplementary Table 1.

**Extrachromosomal assay**

Synthetic DNA for extrachromosomal substrates were purchased from IDT (ultramers; sequences in Supplementary Table 2) and annealed using a thermocycler (Applied Biosystems) in a buffer containing 10 mM Tris-HCL pH 7.5, 100 mM NaCl, and 0.1 mM EDTA. 500 ng of the TMEJ substrate, 20 ng of the NHEJ substrate, and 1 μg of pMAX-GFP (Lonza) were electroporated into 200,000 cells with a single 1350 V, 30 ms pulse using the Neon system (Invitrogen). Cells were pre-treated for two hours with media supplemented with the indicated concentrations of olaparib, ART558, or vehicle (DMSO) prior to electroporation and then incubated in supplemented media again for 30 min post electroporation. Cells were washed with 1x Dulbecco's phosphate-buffered solution (DPBS; Gibco) and then incubated at 37 °C for 10 min in Hank's balanced saline solution containing 25U of Benzonase (Sigma). The QIAamp DNA mini kit (QIAGEN) was used to purify DNA and samples were analyzed using qPCR (Applied Biosystems Quant-Studios 1.7.1) with TaqMan Fast Advanced Master Mix (Applied Biosystems; primers and probes described in Supplementary Table 3). Extrachromosomal TMEJ and NHEJ PCR efficiencies, their independence from each other, and their limits of detection (LOD) were determined by diluting a synthetic DNA containing the extrachromosomal TMEJ or NHEJ amplicon (IDT) into genomic DNA containing constant amounts of the other amplicon sequence (Supplementary Fig. 1a, b). NHEJ joining efficiencies were used to adjust TMEJ efficiencies for each replicate. Experiments consisted of three replicates of the electroporation.

**Cell lines**

All cell lines were cultured at 37 °C and 5% $CO_2$ and confirmed to be free of mycoplasma contamination by PCR (detection limit less than 10 genomes/mL). Mouse embryonic fibroblasts (MEFs) were plated in DMEM (Corning) containing 10% Fetal Bovine Serum (VWR Life

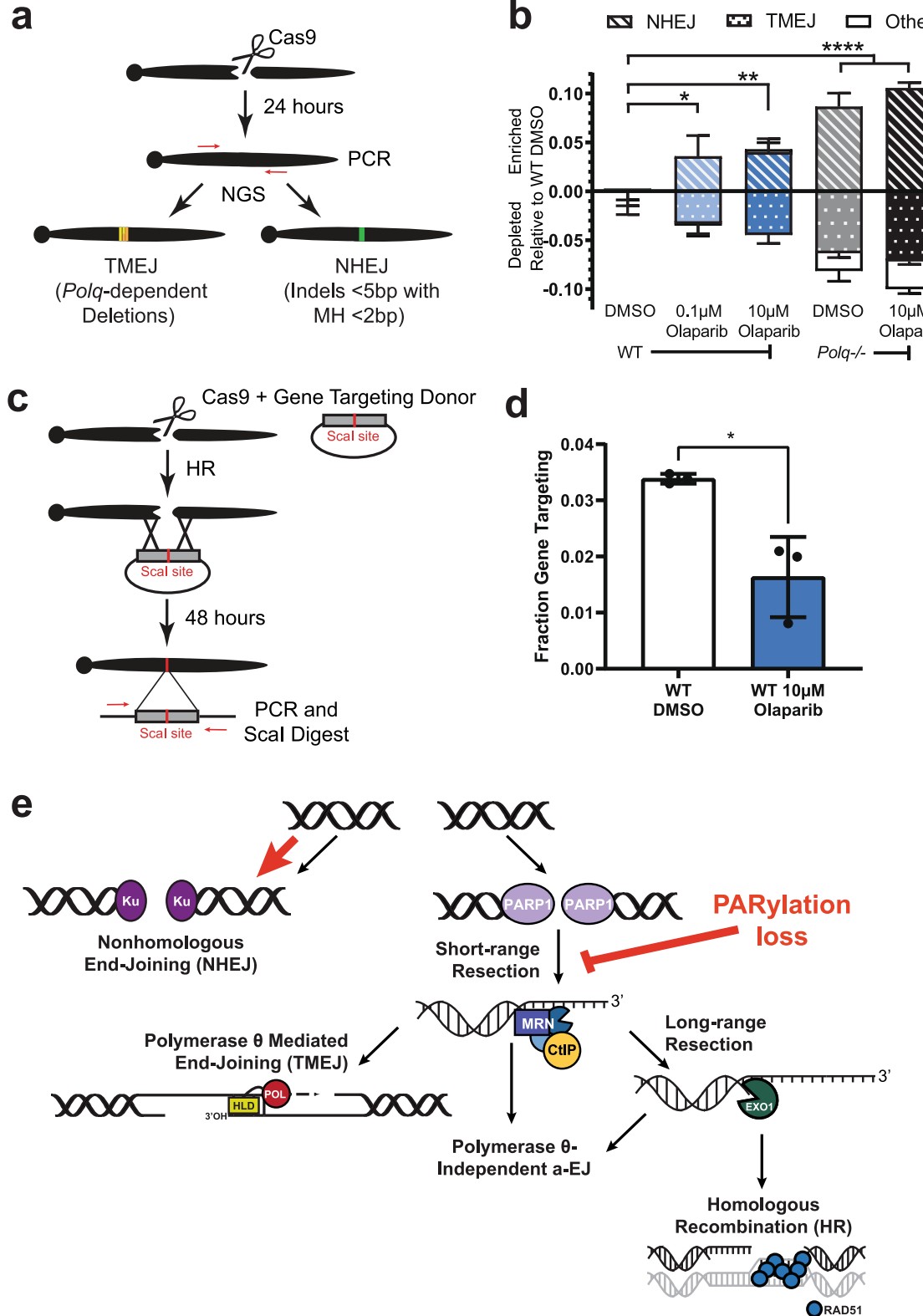

**Fig. 6 | NHEJ compensates for the loss of TMEJ in the absence of PARP activity.**
**a** Amplicon-based, next-generation sequencing (NGS) workflow used to char-acterize all TMEJ and NHEJ repair products. **b** Percent of NHEJ (striped bars), TMEJ (dotted bars), and all other sequences (white bars) enriched or depleted relative to DMSO-treated WT MEFs. Data shown are mean ± SD ($n = 3$). Statistical significance of differences, relative to WT DMSO, refers to NHEJ and TMEJ and was determined by two-way ANOVA (*$p < 0.0017$, **$p < 0.0008$, ****$p < 0.0001$). *Polq-/-* MEFs treated with 10 μM olaparib were the only condition significantly different from WT DMSO for other repair ($p = 0.008$). **c** Diagram depicting gene targeting assay. A donor plasmid containing 550 bp of sequence homology on each side of the Cas9 break was introduced to cells. Repair events using the donor plasmid as a template for repair will include a genomic ScaI site. **d** Fraction of gene targeting in WT MEFs treated with either vehicle or 10 μM olaparib. Data shown are mean ± SD ($n = 3$), and statistical significance of difference determined by unpaired, two-tailed t-test (*$p = 0.049$). **e** Model of the effect of PARylation loss on DSB repair.

## Chromosomal qPCR assays

CRISPR RNAs (crRNAs) specific for sites in the Rosa26 locus are described in Supplementary Table 4. For generating chromosomal DSBs targeted to these sites, 7 pmol of Cas9 was incubated with 8.4 pmol of annealed crRNA+tracrRNA (Alt-R, IDT) at room temperature for 30 mins, then mixed with 32 ng of pMAX-GFP before electroporation of the mixture into 200,000 cells as described above. Two electroporations were pooled for each replicate. Cells were treated with media supplemented with the indicated drug concentrations or DMSO overnight and then plated into media containing fresh drug or DMSO after electroporation. DNA was extracted from cells at indicated time points using a QIAamp DNA mini kit (QIAGEN). Signature repair products were measured via qPCR (Applied Biosystems QuantStudios 1.7.1) using 50 ng of DNA with TaqMan Fast Advanced Master Mix (Applied Biosystems; primers and probes described in Supplementary Table 3). A linear response to decreasing template and LODs were confirmed by diluting DNA from a Cas9-cut, WT DMSO sample recovered 24 (R26$^{MHD}$ and R26 NHEJ) or 48 (R26$^{TINS}$) hours after Cas9 introduction to cells into unbroken genomic DNA (Supplementary Fig. 2a, c and d). All signature qPCRs were normalized to a reference amplicon 32 kb downstream of the Cas9-cut site (Ref2) as measured in an independent PCR reaction.

## Immunoblotting

Whole cell lysates were generated using radioimmunoprecipitation assay (RIPA) buffer with freshly added phosphatase (Sigma, P0044) and protease inhibitors (Sigma, P8340). 50 μg of lysate in sodium dodecyl sulphate (SDS)-loading buffer was loaded onto 8% tris-glycine SDS polyacrylamide gels. Proteins were transferred to a nitrocellulose membrane (General Electric) in tris-glycine transfer buffer containing 20% methanol. Membranes were blocked in 5% fat-free milk (Carnation) for 1 h at room temperature, and then incubated overnight at 4 °C in primary antibody (Supplementary Table 5) diluted in phosphate buffered solution with 0.1% Tween-20 (PBST), and then in LI-COR secondary antibodies (Supplementary Table 5) diluted in PBST for 1 h at room temperature. Membranes were imaged on a Licor Odessey.

## Clonogenic survival assay

KPB13 cells were plated in triplicates for each condition, 1000 cells per 10 cm dish, with 10 ml growth media. Olaparib was added the next day after cell seeding to the final concentrations of 0.1 μM or 0.3 μM. Cells were harvested for crystal violet staining and colony counting seven to ten days later. The clonogenic surviving fraction was determined as the ratio between the plating efficiency of treated versus untreated cells.

## Immunofluorescence

HALO-POLQ expressing RPE-1 cells were plated at 70% density on chamber slides 24 h prior to treatment. Vehicle or olaparib was added immediately prior to irradiation with 10 Gy in a RadSource RS2000 irradiator. Cells recovered for two hours post-irradiation prior to harvest. Janelia Fluor 549 HaloTag Ligand (Promega) was added to the recovery media 15 min prior to collection. Upon collection, cells were fixed with 4% paraformaldehyde (Electron Microscopy Services) and permeabilized with 0.5% Nonidet P-40 substitute (Fluka). The cells were then blocked with 0.5% BSA (Fisher Bioreagents) and 0.2% fish gelatin (Sigma) prior to primary antibody (Supplementary Table 5) incubation in blocking solution overnight. Slides were then washed and incubated with Alexa Fluor secondary antibodies (Supplementary Table 5) in blocking solution. Cells were subsequently washed and stained with DAPI (BioLegend) prior to mounting. Images were acquired on a BX61 Olympus microscopy with recommended Z stack depth optimization and Velocity 6.3 software. Images were processed with Imaris 9.5 and FIJI 2.1.0 software packages.

## Droplet digital PCR

Chromosomal DSBs were induced as described above for the chromosomal qPCR assays using the R26$^{MHD}$ crRNA, and DNA was extracted at the listed time points using QIAamp DNA mini kit (QIAGEN). Samples were digested with NdeI (New England Biolabs; NEB) and either incubated with 50% glycerol (mock-treated; Sigma) or 20 units Thermolabile Exonuclease I (ExoI-treated; NEB) at 37 °C for 2 h in a buffer containing 10 mM Tris-HCl, 50 mM KCl, and 1.5 mM MgCl$_2$. All samples were heated to 65 °C for 5 min to inactivate ExoI. Droplet Digital PCR (ddPCR) was performed with 25 ng of digested DNA and ddPCR Supermix for Probes (no dUTP) (BioRad Laboratories). Droplets were generated and read using QX200 and QX600 AutoDG Droplet Digital PCR Systems and analyzed with QX Manager 1.2 and 2.0, respectively (BioRad Laboratories). TMEJ and NHEJ signatures were measured and normalized using the same primers and probes as described in the chromosomal qPCR assays. We confirmed for the Ref1 and Flank amplicons that amplification was linear in response to the amount of template and independent of multiplexed amplicon amplification efficiency (e.g. Supplementary Fig. 3a, b); we similarly confirmed the "Intact" target amplicon was linear in response to the fraction of broken DNA by serial dilution of XbaI (NEB) digested genomic DNA (XbaI cuts immediate adjacent to Cas9-target site) into unbroken genomic DNA (Supplementary Fig. 3c, green circles). We further confirmed ExoI digestion conditions were specific for ssDNA ends (does not degrade XbaI-generated dsDNA ends; Supplementary Fig. 3c black x's, Supplementary Table 6) and able to fully degrade ssDNA ends using a spike-in ssDNA control (Supplementary Fig. 3d, e, Supplementary Table 6). The fraction of ssDNA (ExoI-sensitive amplification) and deletions >8 bp, >284 bp, and >527 bp were determined using an amplicon ending 7 bp, 283 bp, and 526 bp, respectively, upstream of the Cas9-cut site (flank; Fig. 5a). The minimum length of ssDNA/deletion was determined by the amplification efficiency of synthetic DNAs with progressively increased deletion of the sequence proximal to the break site. The fraction of ssDNA determined by PstI-resistance was completed using DNA digested with 20 units of PstI (NEB) and inactivated for 20 min at 80 °C. The resistant fraction was determined using an amplicon flanking a PstI site 168 nt from the Cas9 break (Flank PstI). The sequence of all primers, probes, and synthetic DNA controls used can be found in Supplementary Table 3.

## Next-generation sequencing library preparation

Chromosomal DSBs and DNA extraction were performed as in the chromosomal assays. Polyacrylamide gel electrophoresis-purified primers (IDT) containing a 6 bp barcode, a spacer sequence of varying length (1–8 bp) to increase library diversity, and 21 (Fwd primer) or 22 (Rev primer) bp of Illumina adapter sequence (sequences in Supplementary Table 7) were used to amplify DNA equivalent to 60,000 genomes for 24 cycles. Libraries were purified with a 2% agarose (Lonza) gel and the QIAquick Gel Extraction Kit (QIAGEN), and then the recovered DNA was amplified for five cycles using secondary NGS PCR primers (Supplementary Table 7) and purified using AMPure XP beads (Beckman Coulter). Libraries were sequenced using an Illumina iSeq 100 i1 kit (300-cycle) with 20% PhiX Control v3 DNA (Illumina).

## High-throughput sequencing junction characterization

The total number of reads generated and analyzed per sample are listed Supplementary Table 8. Data were analyzed using CLC Genomic Workbench 8 as previously described[10] and junctions were characterized using a custom Python script (PyCharm Community Edition 2021, JetBrains) as outlined here. Junctions were scanned for matches of 10 nucleotides, starting proximal to the break site and searching for an upstream and downstream match corresponding to the smallest possible deletion. These matches established deletion length to the left and right, respectively. Junctions were then reconstructed as the sequence

between the 5' end of the upstream match and the 3' end of the downstream match. Junctions were further characterized for insertions (any intervening sequence between the left and right 10 nucleotide matches) and microhomologies (sequence overlap between the left and right 10 nucleotide matches). We excluded junctions containing base ambiguities (i.e. N, W, S, R, K) and junctions with base substitutions in the 3–10 nucleotides proximal to the break site if nucleotides adjacent to the substitution matched the corresponding reference sequence; these substitutions are consistent with polymerase error during sample amplification and are misattributed as insertions. Deletions significantly depleted in *Polq−/−* MEFs identified via the Benjamini-Hochberg procedure[43] were categorized as having significant contributions from TMEJ[10]. In sum, these products accounted for 19% of all repair (Supplementary Table 9), but this fraction was reduced 6.3% by Pol θ deficiency, confirming that these products can also be generated by other repair pathways. We therefore estimated the contribution of TMEJ to repair by similarly subtracting the difference between vehicle-treated WT MEFs and all other conditions. We approximated the contribution of NHEJ to repair by summing the insertions or deletions <5 bp and MH < 2 bp[28] (63% in vehicle-treated WT MEFs). All other sequences (insertions or deletions ≥5 bp, or MH > 2 bp and not significantly depleted by *Polq deficiency;* 17% in vehicle-treated WT MEFs) were listed as "other".

### Gene targeting

A Cas9 break was introduced as described in the chromosomal qPCR assays, except with the inclusion of 1.5 µg of a gene targeting donor plasmid instead of pMAX-GFP per 200,000 cells. After DNA extraction, repair products were amplified using primers outside of the region of homology included on the plasmid to avoid plasmid amplification (Supplementary Table 3; Gene Targeting). PCR products were digested with 20 units of ScaI (NEB) for 2 h at 37 °C. Digestion products were visualized after poly-acrylamide electrophoresis and imaging using a Typhoon FLA 9500. Cy5 bands were quantified using ImageQuant TL Toolbox 8.2, and the fraction of gene targeting was calculated by dividing the intensity of the sensitive band by the sum of the sensitive and resistant band intensities.

### Statistical analysis

The number of replicates and statistical tests are listed with figures and were conducted using GraphPad Prism 9, and all $p$ values determined by ANOVA were corrected for multiple comparisons. Statistical tests were run on cycle threshold ($C_t$) values for all qPCR data, before transformation of data for the linear scale representations in display figures. The Benjamini-Hochberg procedure[43] was used to identify products depleted in *Polq*-deficient cells (Fig. 6b) in Microsoft Excel with a false discovery rate of 10%.

### Reporting summary

Further information on research design is available in the Nature Research Reporting Summary linked to this article.

## Data availability

All raw sequencing data that support the findings of this study have been deposited in the NCBI Sequencing Read Archive (SRA) with the accession code PRJNA806204. All other data supporting the findings of this study are available within the article and its Supplementary Information files. Source data are provided with this paper.

## Code availability

The sequencing analysis code has been deposited in GitHub, https://github.com/aluthman/Ramsden-Lab.git and are available for download at the online repository zenodo.org with the accession code 6799433[44].

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

## Acknowledgements
We would like to thank Artios Pharma Limited for supplying ART-558, Dr. Rick Wood for providing the WT and *Polq*–/– MEFs, Marianna Jones for generating the *Parp1*–/– MEFs, and Megan Happ for her research assistance. This work was supported by the US DOD grant W81XWH18-1-0046, US NIH grants P01CA247773 to D.A.R. and G.P.G., and US grant U01CA097096 to D.A.R. M.E.L. was supported by the US NIGMS grant 5T32 GM007092. S.S. is supported by the US NCI grant F32CA264891. A.J.L. was supported by T32 GM119999.

## Author contributions
M.L. and D.A.R. conceptualized the study. S.S. acquired and analyzed all microscopy. W.F. conducted cell survival assays. A.L. wrote and complied bioinformatics analysis. M.L. generated and analyzed all other data. D.A.R. and G.P.G. supervised the study. M.L. and D.A.R. drafted the manuscript. M.L., S.S., W.F., A.L., G.P.G. and D.A.R. edited and approved the final manuscript.

## Competing interests
G.P.G. receives research funding from Breakpoint Therapeutics, which is developing inhibitors of Polθ. D.A.R. has a materials transfer agreement with Artios Pharma and is using an Artios Pharma compound that inhibits Polθ for research purposes with no financial compensation. M.L. accepted employment at Promega Corporation, manufacturer of the Halo-tag and ligand. The remaining authors declare no competing interests.
