## [Peer Review File · Nature Communications]

Poly(ADP) ribose polymerase promotes DNA polymerase theta-mediated end joining by activation of end resectionREVIEWER COMMENTS

Reviewer #1 (Remarks to the Author):

DNA polymerase theta mediated end joining (TMEJ) is the dominant form of alternative end joining in mammalian cells. While PARP1 has been previously implicated in TMEJ, the extent of its role and how it promotes TMEJ have not been well-described. This study directly addresses these questions. The authors demonstrate that loss or inhibition of PARP activity decreases TMEJ activity several fold, but only on chromosomal substrates that are not pre-resected. Interestingly, the effects on TMEJ of both PARP1 and pol theta inhibition are additive. Indirect immunofluorescence shows PARP inhibition decreases pol theta foci formation and has a small but significant effect on co-localization of pol theta with the end-resection factor CtIP. ddPCR assays are used to highlight a role for PARP activity in promoting TMEJ indirectly through stimulation of short-range resection. These findings are corroborated by the observation of increased NHEJ repair in cells treated with a PARP inhibitor.

Overall, the study is well-designed and addressed an important question with direct relevance to cancer chemotherapeutic approaches. I find the data showing the effects of PARP inhibition on TMEJ efficiency to be solid. I'm less convinced by the experiments showing an effect of PARP inhibition on short-range resection, even though they provide an attractive explanation for the observed TMEJ impairment. While the ddPCR assays are well-controlled, several other contradictory studies have identified a role for PARylation in the prevention of resection at breaks. Therefore, I think the bar is higher here. I encourage the authors to validate their findings related to resection using one or more of the following approaches:

1. Using a second quantification of ssDNA, such as RPA accumulation or loss of restriction endonuclease sites as described by Caron et al. (doi.org/10.1038/s41467-019-10741-9).
2. Knockdown or inhibition of CtIP or MRN activity to validate the ability of the ddPCR assay to accurately measure short-range resection defects.
3. Quantification of the effects of PARP inhibition on the accumulation of intermediate lengths of ssDNA (between 8 and 284 nt), especially given that TMEJ often uses microhomologies found 10-15 nt distal from the break site (Carvajal-Garcia et al., doi.org/10.1073/pnas.1921791117).

Other considerations:

4. The authors consistently use Olaparib as a PARP inhibitor, but as they point out Olaparib can also lead to PARP trapping. They might consider confirming the TMEJ inhibition and inhibition of short-range resection using veliparib, which results in significantly less PARP trapping (Murai et al., [doi: 10.1158/0008-5472.CAN-12-2753](https://doi.org/10.1158/0008-5472.CAN-12-2753)).
5. Since homologous recombination repair requires short-range resection prior to the downstream steps of RAD51 loading and strand invasion, PARP inhibition should also inhibit HR repair, as alluded to in lines 227-228. Is this the case?
6. Line 154: The claim that 'over half of these ends are consistent with "short-range" resection (10-100 nt long ssDNA tails)' seems to be made based on the percentage of ssDNA at distances 284-527 nt from the break. I'm not sure how the authors arrive at the numbers 10-100 nt. Please clarify this logic.
7. Line 160: Should 'supplementary Fig. 4c' actually be 5c?
8. Lines 210-211 and Figure 5B: Is the depletion of the 'other events' by olaparib in the polq -/- background statistically significant, or is this just a trend?

9. Figure 1A: What do the shapes on the DNA in the TMEJ diagram represent? Pol theta helicase and polymerase domains? Or other proteins?

10. Figure 4A: The diagrams here are difficult to interpret. I recommend including the companion diagram from Supplementary Figure 5A as part of the main figure, to make it easy for readers to determine how ddPCR results and subsequent calculations were used to measure resection and the efficiency of different repair types.

11. Supplementary Figure 4 legend: change to 'Example 2D-plot of the ssDNA control and Ref1 multiplexed-ddPCR in mock (D) and ExoI-treated (E) samples.'

Reviewer #2 (Remarks to the Author):

Historically, it has been appreciated that two major DNA double strand break repair pathways (non-homologous end joining and homologous recombination) function to resolve DSBs in virtually all living organisms. That dogma has been revisited since in the last decade or more it has become more and more apparent that an alternative form of NHEJ, collectively termed alternative end-joining, is particularly active if cells are deficient in the NHEJ pathway. However, an emerging consensus agrees that a-EJ can function even in the presence of NHEJ. Moreover, there are likely several forms of a-EJ. A major contributor to one of the a-EJ pathways is DNA polymerase Q, and these authors (and others) now term this form of a-EJ, theta mediated end-joining (TMEJ). What is really not-understood in the field, is how the various sub-types of a-EJ function. In this manuscript from Ramsden and colleagues, a series of elegant end-joining assays are utilized to dissect the impact of PARP1 (and PARP2) activity on end-joining mediated by DNA polymerase theta.

The major finding of the paper is that PARP1 (and its backup PARP2) can enhance TMEJ at chromosomal DSBs, but not on episomal DSBs with pre-resected ends. The authors show nicely that this is likely explained by recruitment of polQ and CtIP to chromosomal DSBs by PARP action at the site of damage. The reduction of TMEJ in cells treated with PARP inhibitors (or that have PARP ablation) is best explained by decreased resection that is requisite for TMEJ. This is a nice contribution to the field.

There are many strengths of this manuscript, including very clever assays that are meticulously validated, a rigorous approach, and addressing a question of considerable interest: "how do different sub-pathways of a-EJ contribute to joining?" Figures 1-3 are compelling. Figure 1 shows that parp inhibitors have zero impact on TMEJ of transfected substrate, whereas a polQ inhibitor (or knockout) completely ablates TMEJ. In contrast, figure 2 shows that PARP inhibitors or ablation partially block TMEJ of chromosomal DSBs.

The authors argue in the introduction that combination therapies targeting both polQ and PARP may be more effective than either alone, but no data is shown in the main manuscript showing this potential effect; but is instead relegated to supplemental data. Supplemental figure 3 should be included in the main manuscript.

In Figure 3, the authors examined polQ and CtIP recruitment to DSBs in the presence of PARP inhibitors and found reduced recruitment of both. Thus PARP's impact on chromosomal TMEJ could potentially be explained by enhancing recruitment of polQ to DSBs, or enhanced resection by CtIP, important because TMEJ requires a resected 3' overhang (figure 3).

Figure 4 assesses the impact of PARP inhibition on end resection at a defined DSB in the ROSA locus. Although the description of the assay in the text provides sufficient information to understand this new assay, the cartoon provided needs to be improved. Still, the data strongly support a critical role for

PARP in promoting short-range resection that is required for TMEJ. Please improve the cartoon depicting this nice assay.

Figure 5 used amplicon sequencing to characterize joints at this DSB in the ROSA locus. This analysis demonstrates a clear shift away from use of microhomology in the absence of polq. The data is entirely presented as histograms, which are useful but not adequate. A table with actual numbers should also be presented in main text. Obviously, assigning TMEJ and NHEJ from sequences is tricky and somewhat arbitrary. It has been long appreciated that NHEJ also utilizes microhomologies. In their normal control, ~19% of the joints are designated as TMEJ mediated (when NHEJ is fully functional, and the ends provided represent a simple DSB). To this reviewer, this seems high. Also, in the normal control, 17% of the sequences were not TMEJ or NHEJ. What are these joints?

Reviewer #3 (Remarks to the Author):

In the manuscript by Luedeman and Ramsden, the authors attempt to decipher the role of PARP enzymes in Pol θ -mediated End-joining (TMEJ) DNA repair. The authors developed extrachromosomal and chromosomal assays to measure TMEJ. Using already pre-resected extrachromosomal substrates, they could not observe any impact of PARP inhibition on TMEJ repair efficiency. Conversely, using Cas9 mediated chromosomal assay they observed a 4-fold reduction of TMEJ upon PARP inhibition either by Olaparib or PARP1/PARP2 depletion. Next, they showed that PARP inhibition impairs DNA end resection and promotes NHEJ. Pol θ localization to CtIP (the TMEJ exonuclease) after damage decreases as well upon PARP inhibition. The authors conclude that PARP enzymes dictate DNA repair choice between alternative and classical end joining by controlling DNA end resection and Pol θ localization to damage sites.

This is an interesting study. Understanding more in depth the role of PARP enzymes in DNA repair is an important question with important translation application notably with regards to the recent development of Pol θ inhibitors. While data are clear and the manuscript is well written, I found the mechanistic insights on the role of PARP in DNA end resection and DNA repair by end joining very preliminary. For instance, data supporting a new function for PARP enzymes in DNA repair are extremely limited in this manuscript. As a result, there is no mechanism on how PARP activity would influence DNA resection. In addition, several panels of the main figures are used to confirm the robustness and efficiency of the TMEJ repair assays developed by the authors. Other panels report data that have already been published by several groups (this is particularly evident for Fig 3).

All in all, my impression is that this study in its current form is too preliminary and does not advance formally the global understanding of how PARP enzymes function in DNA repair.

REVIEWER COMMENTS

I'm less convinced by the experiments showing an effect of PARP inhibition on short-range resection, even though they provide an attractive explanation for the observed TMEJ impairment. While the ddPCR assays are well-controlled, several other contradictory studies have identified a role for PARylation in the prevention of resection at breaks. Therefore, I think the bar is higher here. I encourage the authors to validate their findings related to resection using one or more of the following approaches:

1. Using a second quantification of ssDNA, such as RPA accumulation or loss of restriction endonuclease sites as described by Caron et al. (doi.org/10.1038/s41467-019-10741-9).
2. Knockdown or inhibition of CtIP or MRN activity to validate the ability of the ddPCR assay to accurately measure short-range resection defects.
3. Quantification of the effects of PARP inhibition on the accumulation of intermediate lengths of ssDNA (between 8 and 284 nt), especially given that TMEJ often uses microhomologies found 10-15 nt distal from the break site (Carvajal-Garcia et al., doi.org/10.1073/pnas.1921791117).

As the reviewer notes (and we had previously cited), the literature on the effects of PARP activity on resection had arrived at mixed conclusions prior to our work. We therefore welcome the above suggestions.

- Relevant to suggestion 1, we now also measure resection using loss of amplicon sensitivity to the dsDNA specific restriction enzyme PstI. Our data using this orthogonal method also shows that resection is impaired by PARP inhibitors (new Supplementary Figure 4b). We suggest protein association markers of resection (e.g. RPA localization) are less clearly definitive on this issue. For example, RPA localization could be reduced either because there is less ssDNA, or because it's been replaced by Polθ or Rad51 (see e.g. new work, cited in line 233, where PARylation of RPA promotes its replacement by other factors).
- Relevant to suggestion 2, resection as measured in our assay is reduced in cells deficient in Mre11 (new Supplementary Figure 4a). Anecdotally, we also see increased ExoI sensitive DNA (resected ends) in this assay in cells deficient in 53BP1.
- Relevant to suggestion 3, PstI reports on ssDNA 168nt distal to the break site, providing a distance that is intermediate between the previously measured 8 and 284 nucleotides. We should additionally note that the major product of TMEJ at the locus in question employs a microhomology 6 nucleotides upstream, and 11 nucleotides downstream, from the break site. The observation of differences in resection after PARPi using a

marker 8 nt distal to the break site is thus sufficient to explain differences in TMEJ observed after PARPi.

Other considerations:

4. The authors consistently use Olaparib as a PARP inhibitor, but as they point out Olaparib can also lead to PARP trapping. They might consider confirming the TMEJ inhibition and inhibition of short-range resection using veliparib, which results in significantly less PARP trapping (Murai et al., doi: 10.1158/0008-5472.CAN-12-2753).

We had attempted to address this issue by comparing effects of PARP inhibition (saturating olaparib) to effects of PARP deficiency. As noted in the text of the discussion (starting line 241), we observed modest and not always statistically significant reductions in effect when comparing PARP deficiency vs. PARP inhibition, leading us to suggest a possible mild contribution of trapping. We suggest comparison of the effects of different PARPi is less likely to allow definitive assessment of this possibility.

5. Since homologous recombination repair requires short-range resection prior to the downstream steps of RAD51 loading and strand invasion, PARP inhibition should also inhibit HR repair, as alluded to in lines 227-228. Is this the case?

We assessed repair by HR using a gene targeting assay previously developed by our lab. We now show PARP inhibition indeed suppresses repair by HR (new Figure 6c-d) and have modified text accordingly.

6. Line 154: The claim that ‘over half of these ends are consistent with “short-range” resection (10-100 nt long ssDNA tails)’ seems to be made based on the percentage of ssDNA at distances 284-527 nt from the break. I’m not sure how the authors arrive at the numbers 10-100 nt. Please clarify this logic.

We apologize for the imprecise language. We’ve clarified this paragraph such that only explicit distances are referenced.

7. Line 160: Should ‘supplementary Fig. 4c’ actually be 5c?

This typographical error has been corrected.

8. Lines 210-211 and Figure 5B: Is the depletion of the ‘other events’ by olaparib in the polq -/- background statistically significant, or is this just a trend?

The depletion of other events by olaparib in Polq^{-/-} cells is not statistically significant. It remains possible that olaparib could inhibit Pol θ independent a-EJ, and we continue to note this, but now also make the point that olaparib also has modest effects on HR (Fig. 6c, 6d), consistent with the idea that PARPi’s general effects on resection can impact all resection-dependent pathways.

9. Figure 1A: What do the shapes on the DNA in the TMEJ diagram represent? Pol theta helicase and polymerase domains? Or other proteins?

These shapes do represent the domains of Pol θ . We have edited the figure legend to make this more explicit.

10. Figure 4A: The diagrams here are difficult to interpret. I recommend including the companion diagram from Supplementary Figure 5A as part of the main figure, to make it easy for readers to determine how ddPCR results and subsequent calculations were used to measure resection and the efficiency of different repair types.

The cartoon (now Figure 5) has been modified, and we additionally provide visual cues in subsequent panels to more clearly relate assays to the cartoon.

11. Supplementary Figure 4 legend: change to 'Example 2D-plot of the ssDNA control and Ref1 multiplexed-ddPCR in mock (D) and ExoI-treated (E) samples.'

This typographical error has been corrected.

Reviewer #2 (Remarks to the Author):

The authors argue in the introduction that combination therapies targeting both polQ and PARP may be more effective than either alone, but no data is shown in the main manuscript showing this potential effect; but is instead relegated to supplemental data. Supplemental figure 3 should be included in the main manuscript.

We agree and have transferred this figure to the main manuscript.

Figure 4 assesses the impact of PARP inhibition on end resection at a defined DSB in the ROSA locus. Although the description of the assay in the text provides sufficient information to understand this new assay, the cartoon provided needs to be improved. Still, the data strongly support a critical role for PARP in promoting short-range resection that is required for TMEJ. Please improve the cartoon depicting this nice assay.

As also noted in response to Reviewer 1, the cartoon (now Figure 5) has been modified, and we additionally provide visual cues in subsequent panels to more clearly relate assays to the cartoon.

Figure 5 used amplicon sequencing to characterize joints at this DSB in the ROSA locus. This analysis demonstrates a clear shift away from use of microhomology in the absence of polq. The data is entirely presented as histograms, which are

useful but not adequate. A table with actual numbers should also be presented in main text. Obviously, assigning TMEJ and NHEJ from sequences is tricky and somewhat arbitrary. It has been long appreciated that NHEJ also utilizes microhomologies. In their normal control, ~19% of the joints are designated as TMEJ mediated (when NHEJ is fully functional, and the ends provided represent a simple DSB). To this reviewer, this seems high. Also, in the normal control, 17% of the sequences were not TMEJ or NHEJ. What are these joints?

The statement “19% of the joins are TMEJ” is incorrect, and we thank the reviewer for recognizing this. We defined the subset of products that TMEJ is involved in formally as those significantly (i.e. with false discovery rate < 0.1) depleted in Polq deficient backgrounds. When summed together, this subset accounts for 19% of all products. However, at issue here is the fact that these “TMEJ” products can also be generated by other pathways, including NHEJ. This is the “background” we had referred to in Figure 2A, 2B, and 2C, and it is evident in the new Table requested by the reviewer (13%; Supplementary Table 8). The total amount of TMEJ is thus better defined as the change in product fraction (19%-13%=6%; Figure 6B), and this was our rationale for presenting the data in this manner. We’ve removed the original statement and modified the text to more clearly elaborate our rationale for focusing on the change in product fractions. As suggested by the reviewer we have also added a table summarizing the frequency and standard deviation for the 25 most frequent products.

The 17% of products remaining after applying the definitions for TMEJ and NHEJ described above cannot be assigned to either pathway with any confidence. No single member of this set accounts for more than 1% of all junctions in wild type cells. While they possess little if any microhomology (and are thus more consistent with NHEJ), they have slightly larger deletion than our (as the referee notes) necessarily somewhat arbitrary definition of NHEJ.

Reviewer #3 (Remarks to the Author):

This is an interesting study. Understanding more in depth the role of PARP enzymes in DNA repair is an important question with important translation application notably with regards to the recent development of Polθ inhibitors. While data are clear and the manuscript is well written, I found the mechanistic insights on the role of PARP in DNA end resection and DNA repair by end joining very preliminary. For instance, data supporting a new function for PARP enzymes in DNA repair are extremely limited in this manuscript. As a result, they is no mechanism on how PARP activity would influence DNA resection. In addition, several panels of the main figures are used to confirm the robustness and efficiency of the TMEJ repair assays develop by the authors. Other panels report data that have already been published by several groups (this is particularly evident for Fig 3).

All in all, my impression is that this study in its current form is too preliminary

and does not advance formally the global understanding of how PARP enzymes function in DNA repair.

We agree previously Figure 3A, now Figure 4A, is confirmatory of past work and had noted this in text. It is nevertheless included as a critical preface for the central point of emphasis of this figure, which is Figure 4C. We draw the attention of the reviewer to this second panel; it speaks directly on the role of PARP in end resection, as it shows that PARP inhibition impairs DNA damage dependent colocalization of POLQ and CtIP (CtIP is an initiator of resection). This latter observation has not been reported by others

We have additionally modified text to make it clearer that the central point of this paper is not to describe the role of PARP in DNA repair generally, but rather in Pol theta mediated end joining specifically. As to whether it is preliminary, we respectfully disagree with the reviewer. A role of PARP in a-EJ and TMEJ has long been suggested, but neither the extent of epistasis, nor the mechanism behind this connection, were known. Our revised paper definitively shows that PARP inhibition is not fully epistatic to TMEJ, and we additionally describe the mechanistic basis for this observation. We expect these conclusions are especially of wide interest given the significance of PARPi and Polθi in cancer therapy.

REVIEWERS' COMMENTS

Reviewer #1 (Remarks to the Author):

The authors have included additional experiments and clarified key figures, addressing my main concerns. I am now convinced that loss of PARP1/2 or treatment with PARPi negatively impacts DNA resection, thereby affecting DSB repair pathway choice by inhibiting TMEJ and HR. The addition of the HR targeting experiments nicely completes the story. I think that this carefully designed and controlled study will make a welcome addition to the literature and nicely demonstrates the mechanism by which PARP1/2 promote alternative end joining.

Two suggestions related to Figure 6:

1. The figure legend for 6B reads: "Percent of NHEJ (green outlined), TMEJ (orange outlined), and all other sequences (black outlined)..." I'm not sure what the colors are referring to?
2. Figure 6E shows PARP inhibition preventing short-range inhibition but has a question mark for long-range inhibition. This is inconsistent with what's been changed in the text of the manuscript.

Reviewer #2 (Remarks to the Author):

As in my original review, I believe the authors provide strong evidence that PARP1 enhances TMEJ at chromosomal DSBs, but not on episomal DSBs; and that the impact of PARP ablation is best explained by decreased resection that is requisite for TMEJ. This is a nice contribution to the field.

The authors have in large part, addressed all of my concerns. However, in my original review I suggested that in the current figure 6, the data not be presented solely as histograms, but to also include a table that summarizes the actual percent sequences that are assigned to NHEJ or TMEJ for each experimental condition. The authors say this has been done (supplemental Table8); however, this table is just the top 25 sequences observed in wild type untreated cells.

I still believe a table form depicting the data in Fig.6B (for all experimental conditions) would provide more clarity.

Reviewer #3 (Remarks to the Author):

I understand and agree with the authors that deciphering the role of PARP enzymes in DNA end resection is important. To my view, the authors did not address any of my requests. As stated by the authors themselves: 'This latter observation has not been reported by others', this study describes interesting observations. There is absolutely no mechanism for the proposed role of PARP in end resection. Again, as during my initial reviewing, I found the study interesting but preliminary.

In their response to the reviewer's questions, authors claim they have a mechanism for the role of PARP in end resection. I need an explanation here:

How does PARP enzymes control end resection? Is it through a direct or indirect mechanism? Is it by regulating CtIP, another nuclease, POLQ localization to double strand breaks? Does CtIP binds to PAR chains? Does CtIP knockdown prevent POLQ foci formation?

Authors show that PARPi prevents IR-induced CtIP-POLQ colocalization. How is this related to end resection? Does inhibiting CtIP phenocopy PARPi and prevent IR-mediated POLQ foci formation? Bringing some mechanism insights into the role of PARP1 in regulating end resection will strengthen the current manuscript.

Lastly, authors refer to a recent study, reporting a role for PARP in the unloading of RPA. It is critical they reference Agnel Sfeir's work showing that the POLQ helicase domain can unload RPA and control the balance between NHEJ and a-EJ (Mateos-gomez et al., NSMB 2016, PMID: 29058711). On the same line, the data showing that PARPi prevents HR repair is confusing. Reading the manuscript, it feels like this effect is due to the impairment of end resection. While I understand this is not the main point of their current study, authors should at least reference Livingston work showing that PARP regulates HR through BRCA1_ BARD1 recruitment (Hu et al, Cancer discovery 2014, PMID: 25252691) Authors should discuss these two studies in their discussion.

To note, I don't understand what the title of Figure 3 means: 'PARPi and TMEJ loss have an additive effect'. Can the author explain? I had also a hard time understanding authors response to my questions. For instance, the sentence: Our revised paper definitively shows that PARP inhibition is not fully epistatic to TMEJ. How come an inhibitor be epistatic with a repair pathway ?

Reviewer Comments

Reviewer 1

Two suggestions related to Figure 6:

1. The figure legend for 6B reads: “Percent of NHEJ (green outlined), TMEJ (orange outlined), and all other sequences (black outlined)...” I’m not sure what the colors are referring to?

We have edited the figure legend to reflect the formatting changes in the graph. NHEJ is now represented by striped bars, TMEJ as dotted, and other as white.

2. Figure 6E shows PARP inhibition preventing short-range inhibition but has a question mark for long-range inhibition. This is inconsistent with what’s been changed in the text of the manuscript.

Thank you for bringing this mistake to our attention; we have removed the question mark.

Reviewer 2

The authors have in large part, addressed all of my concerns. However, in my original review I suggested that in the current figure 6, the data not be presented solely as histograms, but to also include a table that summarizes the actual percent sequences that are assigned to NHEJ or TMEJ for each experimental condition. The authors say this has been done (supplemental Table8); however, this table is just the top 25 sequences observed in wild type untreated cells.

I still believe a table form depicting the data in Fig.6B (for all experimental conditions) would provide more clarity.

We apologize for the confusing language used in the legend for now Supplementary Table 9. This table shows the frequency and repair pathway designation for **all** conditions, and are only listed in order of the frequency observed in untreated cells. The legend has been updated to more clearly describe this. The four products determined to be significantly dependent on Polq are included in this table, and all NHEJ and other sequences not listed in this table are <0.25% of repair in all conditions. The full array of sequences analyzed are included in the Source Data file and separated into tabs by their repair classification. We have also made the source raw data free and publicly available as well as the script used to analyze this data.

As described in the revised text, the justification for this experiment is the determination whether repair fates are re-channeled, and this is best represented using the current display figure showing the change in proportion. Summing of repair pathways using a binary assignment does not allow for accurate assessment of this issue, since multiple pathways contribute to many of the observed products.

Reviewer 3

In their response to the reviewer’s questions, authors claim they have a mechanism for the role of PARP in end resection. I need an explanation here:

How does PARP enzymes control end resection? Is it through a direct or indirect mechanism? Is it by regulating CtIP, another nuclease, POLQ localization to double strand breaks? Does CtIP binds to PAR chains? Does CtIP knockdown prevent POLQ foci formation?

Authors show that PARPi prevents IR-induced CtIP-POLQ colocalization. How is this related to end resection? Does inhibiting CtIP phenocopy PARPi and prevent IR-mediated POLQ foci formation?

Bringing some mechanism insights into the role of PARP1 in regulating end resection will strengthen the current manuscript.

Lastly, authors refer to a recent study, reporting a role for PARP in the unloading of RPA. It is critical they

reference Agnel Sfeir's work showing that the POLQ helicase domain can unload RPA and control the balance between NHEJ and a-EJ (Mateos-gomez et al., NSMB 2016, PMID: 29058711). On the same line, the data showing that PARPi prevents HR repair is confusing. Reading the manuscript, it feels like this effect is due to the impairment of end resection. While I understand this is not the main point of their current study, authors should at least reference Livingston work showing that PARP regulates HR through BRCA1_ BARD1 recruitment (Hu et al, Cancer discovery 2014, PMID: 25252691) Authors should discuss these two studies in their discussion.

To note, I don't understand what the title of Figure 3 means: 'PARPi and TMEJ loss have an additive effect'. Can the author explain? I had also a hard time understanding authors response to my questions. For instance, the sentence: Our revised paper definitively shows that PARP inhibition is not fully epistatic to TMEJ. How come an inhibitor be epistatic with a repair pathway?

We have revised our manuscript to ensure readers appreciate mechanistic insight provided by this work does not extend to a determination of how PARylation impinges on resection (line 226); we apologize and had not intended to imply otherwise. We agree this is an interesting issue, and as is highlighted by this reviewer, there are a diverse array of non-mutually exclusive mechanisms by which PARP could affect resection, each of which bears a rigorous investigation in future work. We further proofed the manuscript to ensure we clearly state insight into mechanism provided by our work is focused on identifying the steps in TMEJ that are, and are not, impacted by PARP activity.

Appropriately rigorous analysis of PARP effects on HR must also await future work. The study we presented here, and our current brief discussion of this result, is appropriately limited to addressing the prediction raised by another reviewer. We now additionally cite the Livingstone work.

We agree the Mateos-Gomez reference provided important insight into the inhibitory effect of RPA on TMEJ, and now include a citation to this important work in this revision. We note here that participation of the Pol θ helicase domain in helping unload RPA does not preclude a significant additional contribution from stimulation of RPA unloading by PARylation.

Regarding Figure 3; we had meant this statement to refer only to Figure 3B, where treatment with either inhibitor alone had effect on TMEJ that was intermediate to combined inhibition. We have re-titled this figure. We agree employing the term epistasis when considering factor inhibition is formally incorrect, in that epistasis refers specifically to interactions of a gene (like PARP1) with a genetically defined pathway, and not between gene inhibitors and that pathway. The text in the revised manuscript limits use of the term epistasis to the observation that pathway function, as defined by repair that is reliant on the gene Pol theta, remains robust even after fully ablating PARylation. One of the approaches we used to address this was genetic deficiency in PARP enzymes, suggesting the use of the term epistasis was appropriate.